# Novel Molecular Mechanisms of Gangliosides in the Nervous System Elucidated by Genetic Engineering

**DOI:** 10.3390/ijms21061906

**Published:** 2020-03-11

**Authors:** Koichi Furukawa, Yuhsuke Ohmi, Farhana Yesmin, Orie Tajima, Yuji Kondo, Pu Zhang, Noboru Hashimoto, Yuki Ohkawa, Robiul H. Bhuiyan, Keiko Furukawa

**Affiliations:** 1Department of Biomedical Sciences, Chubu University College of Life and Health Sciences, 1200 Matsumoto, Kasugai, Aichi 487-8501, Japan; farhana7779@gmail.com (F.Y.); oriet@isc.chubu.ac.jp (O.T.); apor0825@yahoo.co.jp (P.Z.); biochemistrobi79@gmail.com (R.H.B.); keikofu@isc.chubu.ac.jp (K.F.); 2Department of Biochemistry II, Nagoya University Graduate School of Medicine, 65 Tsurumai, Showa-ku, Nagoya 466-0065, Japan; Yuji-Kondo@omrf.org; 3Department of Medical Technology, Chubu University College of Life and Health Sciences, 1200 Matsumoto, Kasugai, Aichi 487-8501, Japan; ooumi82@isc.chubu.ac.jp; 4Department of Tissue Regeneration, Tokushima University Graduate School of Biomedical Sciences, 3-18-5, Kuramoto-cho, Tokushima 770-8504, Japan; nhashimoto@tokushima-u.ac.jp; 5Department of Glycooncology, Osaka International Cancer Institute, Osaka 541-8567, Japan; yuki34@mc.pref.osaka.jp

**Keywords:** ganglioside, knockout, neurodegeneration, glycosphingolipid, inflammation, microdomain

## Abstract

Acidic glycosphingolipids, i.e., gangliosides, are predominantly and consistently expressed in nervous tissues of vertebrates at high levels. Therefore, they are considered to be involved in the development and function of nervous systems. Recent studies involving genetic engineering of glycosyltransferase genes have revealed novel aspects of the roles of gangliosides in the regulation of nervous tissues. In this review, novel findings regarding ganglioside functions and their modes of action elucidated mainly by studies of gene knockout mice are summarized. In particular, the roles of gangliosides in the regulation of lipid rafts to maintain the integrity of nervous systems are reported with a focus on the roles in the regulation of neuro-inflammation and neurodegeneration via complement systems. In addition, recent advances in studies of congenital neurological disorders due to genetic mutations of ganglioside synthase genes and also in the techniques for the analysis of ganglioside functions are introduced.

## 1. Introduction

Nervous tissues are differentiated from the ectoderm, and their morphology is determined before birth. During the growth stage after birth, further functional differentiation proceeds, and fundamental shapes and functions are established within several years in the case of humans. However, the formation of functional networks based on various experiences and learning activities proceed continuously, leading to the maintenance of high-grade nerve functions due to the plasticity of nervous systems [1]. After reaching middle age, the network function gradually declines along with aging, and physiological and pathological degeneration gradually proceed in nervous tissues [2]. Under certain pathological conditions, marked tissue degeneration is induced, leading to functionally irreversible states such as the occurrence of dementia. During individual processes in the natural history of nervous systems, structures of carbohydrates on proteins and lipids markedly alter [3] and are involved in the formation of appropriate environments for the high-grade structures and functions based on the molecular modification required for each step.

Since acidic glycosphingolipids, i.e., gangliosides, are predominantly and consistently expressed in nervous tissues of vertebrates at high levels [3], their marked contribution to the neurological function has long been expected [4]. Actually, the fact that ganglioside expression patterns markedly change during development, suggests that they play critical roles in the evolution and differentiation of nervous systems [5,6]. Contrarily, for carbohydrates on proteins, sufficient understanding has been achieved in terms of integrative analysis of nervous system-specific carbohydrate functions, although there have been a number of reports on carbohydrate functions on individual proteins. 

In this review, an outline of functions of gangliosides recently identified is summarized with a focus on findings from studies of knockout mice of various glycosyltransferase genes. 

## 2. Roles of Gangliosides

Gangliosides are sialic acid-containing glycosphingolipids, which are widely expressed in almost all tissues and cells of vertebrates. They are enriched in brain tissues, suggesting that they are involved in the evolution and regulation of nervous systems [7]. The most intriguing feature of gangliosides is that they consist of hydrophilic carbohydrates and hydrophobic lipid portions [8], being expressed on the cell membrane and present in the outer layer of the lipid bilayer, as shown in Figure 1. 

Therefore, it has remained unclear how gangliosides are involved in the regulation of signals for cell differentiation, activation, and malignant transformation [9]. This review summarizes ganglioside expression in inflammation and neurodegeneration and their roles in the maintenance of integrity and generation of phenotypes of cells with a focus on inflammation and degeneration due to the altered gangliosides are also summarized.

### 2.1. Gangliosides in Development and Growth

The fact that ganglioside composition in brain tissues varies along with development and growth of organisms has been well reported [7,11]. It is also well understood that simple structure gangliosides such as GM3 and GD3 mainly exist at the initial stage of development, i.e., embryonal day 12 to 14 (E12–14) of mice. At the differentiation stage after E16, when extension of neurites and synapse formation occur, mature-type gangliosides such as GM1, GD1a, GD1b, and GT1b increase and become the main components of brain tissues [12]. The main pathway of ganglioside synthesis is shown in Figure 2. Generally, a-series and b-series gangliosides are major structures in brain tissues, and minimal levels of c-series and asialo-series are sometimes detected. In particular, complex gangliosides containing extended core structures higher than GM1 and GD1b are frequently expressed in brain tissues after the late developmental stage, and sometimes affect pathological conditions such as Alzheimer’s disease, as described below. 

Differently, the ganglioside composition in brain tissues shows almost no change in the adult stage, while total amounts gradually reduce. In particular sites of the brain, however, gradual changes occur along with aging [7]. Only a few comprehensive studies on ganglioside expression in nerve tissues have been performed to date [13,14]. 

### 2.2. Function of Monosialylgangliosides and Disialylgangliosides

In order to analyze the regulatory function of nervous systems by gangliosides, a rat pheochromocytoma cell line, PC12, has been frequently used [15,16]. On the one hand, PC12 cells overexpressing GM1 showed reduced sensitivity to the nerve growth factor (NGF), leading to lowered neurite extension, and exhibited suppressed activation of TrkA/Ras/ERK1/2 signals upon NGF stimulation [17]. On the other hand, GD3-overexpressing PC12 cells showed increased phosphorylation levels of TrkA and ERK1/2 even without NGF stimulation [18]. Similarly, contrastive effects of gene expression between GM1 synthase and GD3 synthase on phenotypes of various cancer cell lines have been observed [19,20]. From these results, it has been demonstrated that monosialylgangliosides and disialylgangliosides play distinct roles in the regulation of malignant properties of cancer cells [9].

Although mechanisms for signal regulation in cancer cells and PC12 cells based on gangliosides with different numbers of sialic acids remain unclear, it could be a reflection of regulatory functions of gangliosides in organogenesis and differentiation of nervous tissues and cells. Dynamic changes in ganglioside expression during the evolution and development of nervous tissues and their implications are summarized in the next chapter. 

## 3. Aging, Neurodegeneration, and Gangliosides

In various tissues and cells, only nervous tissues have been thought to show consistent compositions among species of mammals and birds. In fact, very similar thin-layer chromatography (TLC) patterns of gangliosides extracted from brain tissues were reported [21]. However, various changes in ganglioside composition under physiological and pathological conditions have been investigated [2,8,22]. 

### 3.1. Changes of Gangliosides in Central Nervous Systems with Aging 

There have been some reports on the changes of ganglioside expression during evolution, development, and aging of mice and rats [23,24,25]. For human brain gangliosides, changes of ganglioside expression have been reported. Generally, contents of gangliosides gradually decrease [26], and, on the one hand, a-series gangliosides tend to decrease mainly in the frontal cortex [27]. On the other hand, b-series gangliosides decrease in the cerebellum with aging. Total amounts of lipids contained in brain tissues continuously reduce until reaching 90 years old. In particular, expression levels of gangliosides and cerebrosides are largely lowered at that time point. Investigation of 118 individuals aged 20 to approximately 100 years old revealed that concentrations of gangliosides were maintained between 20 and approximately 70 years old [28], and then the expression patterns continuously altered with aging, leading to reduced ratios of GM1 and GD1a. Similar findings were reported based on the newest analytical techniques using mass spectrometry [29]. However, there has been no detailed research on the functional relationship of altered ganglioside expression with pathological changes in brain tissues. 

### 3.2. Gangliosides in Alzheimer’s Disease

In a variety of neurodegenerative diseases, particularly Alzheimer’s disease, one of the most important factors involved in neuron death is local inflammation [30]. As reported previously, all components involved in the classic pathway of complement activation were detected in nervous tissues, and this pathway is activated in Alzheimer’s disease, resulting in detection as fibrous β amyloid [30] or neurofibrillary tangles [31]. From these findings, complement activation and subsequent inflammation can be the main mechanisms for the induction of brain damage in Alzheimer’s disease.

Over approximately the last 20 to 30 years, it has been demonstrated that the complement system is involved in neurodegeneration such as in Alzheimer’s disease [32]. C1q and other components deposit in amyloid plaques and neurofibrillary tangles [33,34], leading to complement activation. In areas with legions, levels of mRNAs for complement components markedly increased [35]. Furthermore, C1q inhibitors alleviated clinical features of Alzheimer’s disease, suggesting that complement activation plays important roles in neuroinflammation and subsequent neurodegeneration [36]. Interestingly, the complement activation observed in double knockout (DKO) mice (deleting GM2/GD2 synthase and GD3 synthase) containing only GM3 plays similar roles to those in Alzheimer’s disease. This point will be described later.

Furthermore, complement systems can have both detrimental and beneficial effects regarding disease control [37]. For example, roles during developmental processes [38] or neuronal generation in adults [39] have been reported. In particular, its physiological roles in the elimination of unnecessary cellular components or in the improvement of inflammatory reaction are of interest. 

As a role of gangliosides in Alzheimer’s disease, GM1 has been reported to be a factor triggering the aggregation of Aβ peptides on the cell membrane [40], and mechanisms leading to the continuous accumulation of Aβ have also been proposed [41]. Murine models of Alzheimer’s disease generated with genetic backgrounds of knockout of the GD3 synthase gene or GM2/GD2 synthase gene have been generated, resulting in milder phenotypes in the former [42], and more serious phenotypes in the latter [43]. Recently, it was reported that the expression of *B4GALNT1* promoted the processing of Aβ [44], and complex gangliosides exacerbated clinical features of Alzheimer’s disease. Additionally, different changes in ganglioside composition were found in various transgenic model mice expressing human amyloid precursor proteins. [45]. From these results, it has been suggested that altered metabolism of gangliosides is involved in the pathogenesis of Alzheimer’s disease. 

### 3.3. Parkinson Disease and Gangliosides

It has been reported that the amounts of GM1 and GD1a are reduced in the brains of patients with Parkinson disease [46]. Furthermore, it was also indicated that *B4galnt1* (GM2 synthase) KO mice lacking GM1 ganglioside [47] showed Parkinson disease-like neurological disorders even in heterozygotes [48]. Indeed, the efficiency of GM1 administration to Parkinson disease model animals has been reported [49], and similar trials have also been performed in clinical cases, resulting in the improvement of clinical features [50]. For example, GD3 synthase9 (*ST8SIA1*) KO mice lacking b-series gangliosides and showing increased levels of GM1 and GD1a [51] were reported to be less susceptible to Parkinson disease [52], suggesting that supplementation of GM1 or GD1a is also effective for patients with other neurodegenerative diseases [46]. 

### 3.4. Inflammatory Reaction and Gangliosides

Changes in ganglioside expression in various inflammatory reactions have been observed to date [53]. For example, changes in gangliosides expressed on glial cells [54] or in multiple sclerosis, or changes by growth factors [55] have been reported. Furthermore, it has been reported that the ganglioside GD3 was induced in inflammatory environments of brain tissues in mice and rats [56]. Regarding immune cells, GD3/GD2 were induced in T lymphocytes when stimulated via T-cell receptors or IL-2 receptors, or various mitogenic factors [57,58]. Moreover, ganglioside GD2 was expressed on functional T cells [59]. These results are consistent with a report that GD2 was induced on HTLV-1-infected T cells via a viral product, p40tax [60]. Thus, changes in ganglioside expression patterns on immune cells are frequently observed in various inflammatory reactions [61,62].

## 4. Functions of Glycolipids Elucidated in Ganglioside-Deficient (Knockout) Mice

### 4.1. Abnormal Phenotypes Exhibited by Knockout Mice and Inflammatory Reaction 

In order to investigate roles of glycosphingolipids, cDNAs of various glycosyltransferase genes have been isolated, i.e., GM2/GD2 synthase [63], GD3 synthase [64,65,66], GM1/GD1b/GA1 synthase [67], GM3 synthase [68,69,70], Gb3 synthase [71,72,73], and many other glycosyltransferase cDNAs, and then knockout (KO) mouse lines of these enzyme genes have been established. For example, KO mouse lines of GM2/GD2 synthase [47], GD3 synthase [51], Gb3 synthase [74], lactosylceramide (LacCer) synthase [75,76], and double KO mice of GM2/GD2 synthase and GD3 synthase [77,78] have been established and analyzed. DKO mice of *B6galt5* and *B4galt6* were also generated [79]. Synthetic pathways of the main glycosphingolipids and glycolipid structures deleted in individual KO mice are presented in Figure 2. Generally, abnormal phenotypes observed in these KO mouse lines were milder than expected. This could be because residual glycolipids compensate for the functions of deleted structures [47]. All these results are summarized in Table 1. However, it is interesting that inflammatory reactions were found mainly in the central nervous systems of many of these KO mice involving glycosyltransferase genes [80]. Biochemical and morphological changes observed in the DKO mice of GM2/GD2 synthase and GD3 synthase genes were also detected in many single gene KO mouse lines [81] (Table 1).

We reported the involvement of complement systems in neuro-inflammation as a novel aspect of ganglioside deficiency [80]. We compared the degree of complement activation, inflammatory reaction, and destruction of lipid rafts among various KO mice of glycosyltransferase genes and wild type (WT) mice and demonstrated extensively increased expression levels of complement-related genes. Moreover, we reported the proliferation of astrocytes and assembly of microglia corresponding to the degree of defects in ganglioside composition in individual KO mice (Figure 3). It was also shown that various cytokine genes were upregulated with aging, corresponding with the progression of neuro-inflammation, as described above. The molecular mechanisms of this inflammation based on ganglioside deficiency were analyzed with a focus on changes in lipid rafts [81]. Details are described in the Section 5. 

### 4.2. Neuro-Inflammation Corresponding to the Degree of Ganglioside Deficiency

Gangliosides have been considered to be involved in the development, differentiation, and function of nervous systems [82]. However, gangliosides have been shown to play roles mainly in the maintenance and repair of nervous tissues based on the abnormal phenotypes detected in genetically engineered mutant mice [83]. 

Generally, neurodegeneration was commonly found in KO mouse lines of ganglioside synthase genes [47,84,85]. In particular, age-dependent progressive neurodegeneration was observed in KO mice of GM2/GD2 synthase, while subtle abnormal neurological signs could be detected when they were born [47]. In addition, DKO mice of GM2/GD2 synthase and GD3 synthase genes demonstrated neurodegeneration in the early stage of life [77], or even sudden death by auditory stimulation [86]. Although KO mice lacking GlcCer synthase [87] showed embryonal lethality [88], conditional KO mice in which GlcCer synthase was deleted after birth also showed neurodegeneration [36]. These results indicate that ganglioside deficiency causes abnormality in the maintenance of integrity of the nervous system, leading to neurodegeneration. However, it remains unclear how ganglioside deficiency causes neurodegeneration. 

Among various features indicating inflammation in the nervous tissues, abnormal proliferation of astrocytes and assembly of microglia were markedly and characteristically found in the cerebella of ganglioside deficient mice [80]. Furthermore, these inflammatory reactions were confirmed by immunohistochemistry, such as GFAP-positive astrocytes and F4/80 antibody-reactive microglia (Figure 3). GFAP+ cells increased at 15 weeks after birth in DKO mice, and further increased with aging. At 50 weeks after birth, GFAP+ cells increased even in single gene KO mice, such as GD3 synthase KO or GM2/GD2 synthase KO. Microglia cells also showed increased assembly at 15 weeks after birth in DKO mice, and further increased with aging. This microglia assembly could also be found at 50 weeks after birth in single gene KO mice [80]. 

As for inflammatory cytokines, increased expression levels of IL-1β and TNFα genes were detected in RT-PCR of mRNA from the cerebella of DKO mice. Expression levels of these genes tended to increase with aging in DKO mice, while no apparent changes in those gene expression could be found in WT or single gene KO mice. 

### 4.3. Involvement of Complement System in the Inflammatory Reaction 

From the results of gene expression analysis in DKO mice, it was demonstrated that complement-related genes were generally upregulated in the cerebella of DKO mice. Therefore, it was suggested that wide-ranging consumption of complement components was induced due to activation of the complement system [80]. Actually, the deposition of C1q, a complement component, could be found in the cerebella of DKO mice, and it was also the case in KO mice of GM2/GD2 synthase. [80]. In order to investigate whether complement activation detected in the cerebella of DKO mice exacerbates with aging, expression levels of C1qα, C3, and C4 were examined along with aging, resulting in increased expression of the C1q gene between 15 weeks to 50 weeks after birth, and in differences from WT mice with aging. In GM2/GD2 synthase KO mice, expression of complement-related genes moderately increased. Similar inflammatory reactions were also observed in spinal cords of these KO mice [89].

To clarify the roles of complement activation in neuro-inflammation and neurodegeneration, triple KO mice lacking the complement C3 gene, as well as GM2/GD2 synthase and GD3 synthase genes, were established as shown in Figure 4 [83]. In these TKO mice, it was shown that complement activation is involved in complement deposition and secretion of inflammatory cytokines and also in neurodegeneration, as demonstrated by the alleviation of brain degeneration indicated by a reduction in brain weights. In neurological disorders such as Guillan–Barre syndrome and Miller syndrome caused by anti-ganglioside antibodies, it has been reported that complement systems are closely involved [90,91]. Therefore, inhibitors of complement-related components have undergone therapeutic application for the control of these diseases, showing beneficial effects in mouse disease models [92] and human cases [93]. 

## 5. Microdomains on Cell Membrane and Gangliosides 

Generally speaking, membrane microdomains such as lipid rafts, glycolipid-enriched microdomain (GEM)/rafts, or detergent-insoluble microdomains (DIM) are considered to be a platform of cell signaling, and roles of glycosphingolipids in lipid rafts have been increasingly recognized [94]. In particular, the molecular composition of gangliosides that consist of polymorphic sugar chains and heterogenous lipid moieties has led to the expectation that gangliosides could be one of the main regulators of biological properties of microdomains.

### 5.1. Gangliosides Regulate Cell Signaling in Microdomains 

Various extrinsic stimulations are transduced via receptors and their adjacent molecules on the cell membrane, and these molecules often form molecular complexes in microdomains, such as lipid rafts or GEM/rafts [9,95] (Figure 5). With a number of experiments using cell lines and KO mice, it has been shown that changes in ganglioside expression largely affect lipid rafts and control the cell signals, and finally cellular phenotypes [94]. Therefore, the integrity of lipid rafts has been investigated by analyzing changes in the intracellular localization of membrane molecules depending on the conditions of glycosphingolipids and cells [96,97]. Immunocytostaining of these membrane molecules has been an efficient approach for the localization of the same membrane microdomains. Furthermore, it seems extremely important to substantially reveal physical interactions among these molecules on the living cell membrane, which would become a prerequisite to clarify the roles of gangliosides and their associated molecules on the cell membrane [98].

### 5.2. Microdomain on the Cell Membrane of Nervous Systems and Gangliosides 

Glycosphingolipids are generally considered to be concentrated and localized in GEM/rafts [99], and intense localization is found in highly differentiated neurons. However, they are dispersed from lipid rafts in particular environments [100]. In addition, changes in relevant molecules such as caveolin-1 affect the intracellular localization of glycosphingolipids [101]. Furthermore, fine distribution analysis of gangliosides using immunoelectron microscopy revealed that different ganglioside species (e.g., GM1 and GM3) showed distinct distribution patterns on the cell membrane, suggesting the presence of heterogeneous microdomains, and that individual gangliosides form specific microdomains [102]. 

Surprisingly, GPI-anchored proteins and GEM/raft markers dispersed from GEM/rafts in the cerebella of ganglioside-deficient mice, and marked cell damage was induced [80,81,103]. Analysis of altered floating patterns of GPI-anchored proteins and GEM/rafts markers in various ganglioside-deficient mouse lines by immunoblotting revealed that flotillin-1 and caveolin-1 were dispersed from GEM/rafts (Figure 5). DAF and NCAM showed marked dispersion from GEM/rafts in DKO mice, and DAF shifted to the non-GEM/raft fraction in GM2/GD2 synthase KO mice. Total protein amounts showed no differences among these lines. Generally, gangliosides are essential in the architecture of GEM/rafts, and it is suggested that more marked abnormalities of GEM/rafts are exhibited in DKO mice than in single gene KO mice. 

### 5.3. Complement Activation and Destruction of Lipid Rafts

The results of the analysis of DKO mice revealed that there were many examples of molecular dispersion of GPI-anchored proteins and GEM/raft markers even in single gene KO mice, suggesting that disordered GEM/rafts in brain tissues induce dysfunction of GPI-anchored proteins. In GM2/GD2 synthase gene KO and DKO mice, localization of GalCer, phospholipids, and cholesterol tended to decrease [81], suggesting that the abnormal composition of gangliosides could induce abnormal distribution patterns of GPI-anchored proteins and raft marker proteins, and also of other lipids.

One of the most important factors involved in the complement activation and resulting neurological disorders should be complement-regulatory proteins. In fact, it is well known that expression levels of CD59 are lower at the sites of Alzheimer’s disease [104]. DAF is also a crucial molecule for the maintenance of tissue integrity [105]. Many of these complement-regulatory factors belong to GPI-anchored proteins and are concentrated in the GEM/raft fraction [106]. Therefore, it is suggested that destruction of the GEM/raft induced changes in the localization of GPI-anchored proteins and their functional abnormalities, leading to complement activation and inflammation. These processes are summarized in Figure 5.

## 6. Human Diseases Caused by Congenital Deficiency of Gangliosides

Following the analysis of ganglioside functions using KO mice of glycosyltransferase genes, human cases of congenital defects of ganglioside synthase genes have been reported in this century. 

### 6.1. GM3 Synthase Deficiency Causes Severe Clinical Features

Although there have been a number of studies on deficiency of ganglioside catalytic enzyme genes, no reports on congenital deficiency of ganglioside synthase genes were published until 2004. Simpson et al. reported “infantile epilepsy” in Amish families due to deficiency of GM3 synthase (*ST3GAL5*) as the first case of genetic mutation in ganglioside synthase genes [107]. 

As described above, the majority of gangliosides are synthesized through GM3, and diverse carbohydrate structures are generated from a common precursor, lactosylceramide, along with several major synthetic pathways [25]. Therefore, defects of the GM3 synthase gene in the Amish families actually resulted in the complete loss of ganglio series and the patients exhibited serious infantile epileptic disorders [107] and skin abnormalities [108,109], suggesting that gangliosides are essential in the regulation of nervous tissues and other organs. Thus, patients lacking GM3 synthase activity exhibited severe neurological disorders such as infantile epilepsy, mental retardation, visual disorders, and also skin pigmentation abnormalities, while no definite abnormal phenotypes were found in KO mice of the GM3 synthase gene (*St3gal5*) in any sites of the body except for the auditory system [110].

### 6.2. GM2/GD2 Synthase Gene Deficiency Causes Hereditary Spastic Paraplegia 

B4GALNT1 is an essential enzyme for the synthesis of complex gangliosides, the lack of which resulted in progressive neurodegeneration with aging in mice [84]. Recently, 11 cases of hereditary spastic paraplegia (HSP) due to mutation in the coding region of *B4GALNT1* were reported [111,112,113]. We examined the enzyme activities using a cell-free enzyme assay with cell extracts, and by flow cytometry of transfected cells with mutant cDNA expression plasmids [114]. Among them, almost all mutant genes showed the complete loss of B4GALNT1 activity, while two mutants showed low activity, indicating that the clinical findings of these patients were derived from the loss of B4GALNT1 enzyme activity, and that mutations were responsible for the clinical features of HSP. As expected from KO mice phenotypes of the *B4galnt1* gene, the intensity of their neurological disorders was milder than expected. These clinical features of patients including male hypogonadism are very similar to the abnormal phenotypes detected in *B4galnt1*-deficient mice [115]. In contrast to GM3 synthase mutation, *B4GALNT1* mutations brought about much milder clinical features with slower progression.

## 7. Future Scope of Ganglioside Research

Recent advances in methodologies and technologies have enabled us to further investigate modes of action of gangliosides. Fine heterogeneity in either the sugar moiety or ceramide portion has been demonstrated, leading to further understanding of the mechanisms by which gangliosides play their roles or interact with their recognizing molecules. For example, further analysis of derivatives of sialic acids such as deaminoneuraminic acid (KDN) [116], *O*-acetylated GD3/GD2 [117] as sugar modifications, and long fatty chain-containing glycolipids in lactosylceramide [118], saturated/unsaturated fatty acid-containing Gb4 [119], or hydroxylated ceramide-containing gangliosides [120], remain to be promoted. It is also important to identify the cell lineages that more critically need gangliosides between neurons and glia [121], although few studies have been reported to date. Ultrahigh-resolution imaging of gangliosides in GEM/rafts has enabled us to understand the actual formation of microdomains [122] and to generate new concepts regarding the gradual formation of GEM/rafts with different sizes and compositions [123]. Identification of novel ligand molecules for gangliosides should also be promoted to further understand the molecular functions of gangliosides [98], although few studies in the neurology field have been reported to date.

## 8. Conclusions

It is difficult to clearly answer the fundamental question, i.e., “What are the roles of gangliosides in nerve functions?” Many of the “functions” presented here are results drawn from observation of abnormal situations brought about by artificially enhanced expression of particular glycosyltransferases or suppression of their functions to speculate on the normal functions primarily exerted. Therefore, they represent a part of “functions”, but not all. The main limiting factor is the technical restriction in the manipulation of key glycosyltransferase genes, observing some phenotypes as functions of many structures belonging to one group all together. However, clarification of the roles of individual gangliosides by collecting experimental data with various limitations is essential to understand the polymorphism of carbohydrates.

## Figures and Tables

**Figure 1 ijms-21-01906-f001:**
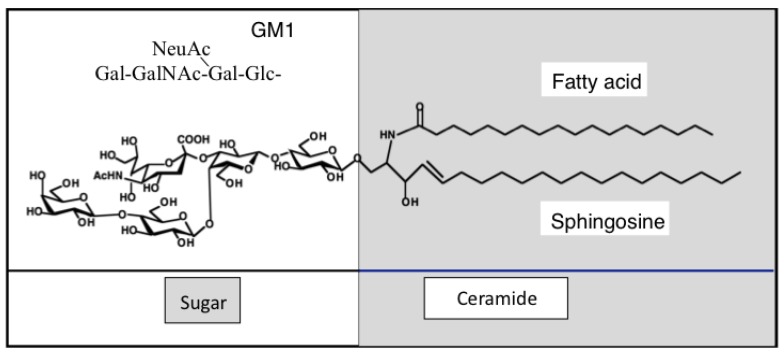
Glycosphingolipids are amphipathic molecules, expressed in the out layer of lipid two layers of the cell membrane. The hydrophobic portion (ceramide) is embedded in the outer layer of the membrane, and the sugar portion is protruding outside the membrane. A representative GM1 structure is shown. Ganglioside nomenclature is according to Svennerholm [10].

**Figure 2 ijms-21-01906-f002:**
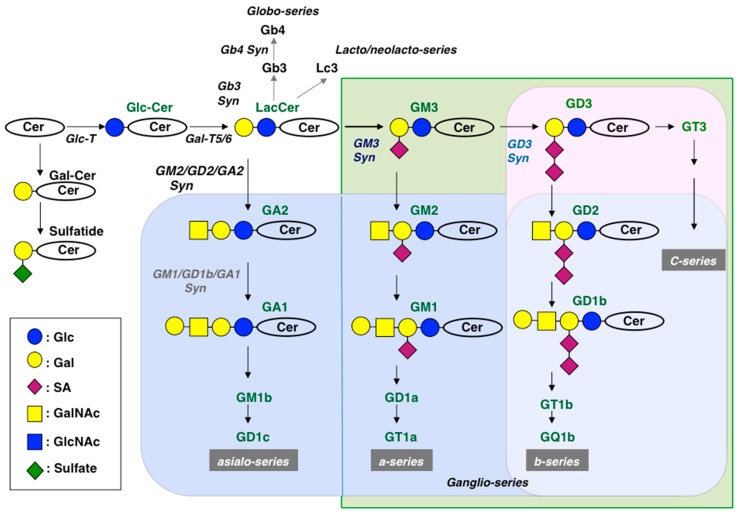
Main pathway of ganglioside synthesis. Glycosyltransferases catalyzing individual steps are shown by italics, and deleted structures in KO mice are shown by different colored squares. Structures on the line of ganglioside synthesis are presented in green letters.

**Figure 3 ijms-21-01906-f003:**
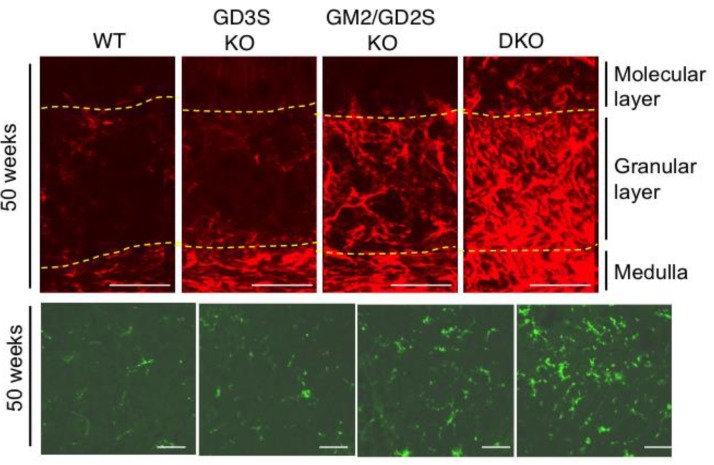
Immunohistochemistry of GFAP-positive astrocytes and F4/80-positive microglia. Cerebella from 50-week-old mice were analyzed by anti-glial fibrillary acidic protein (GFAP) antibody and Alexa Fluor 555-conjugated anti-mouse IgG1, and anti-mouse F4/80 and Alexa Fluor 488-conjugated anti-rat IgG. Marked gliosis was found in ganglioside synthase gene KO mice. GFAP-positive cells were increased (upper, red), and F4/80-positive cells (lower, green) accumulated in cerebella, indicating astrocytes and microglia, respectively. Scale bar, 50 µm in both panels. Borders between layers of the cerebellum indicated based on Hematoxylin-Eosin staining of serial sections.

**Figure 4 ijms-21-01906-f004:**
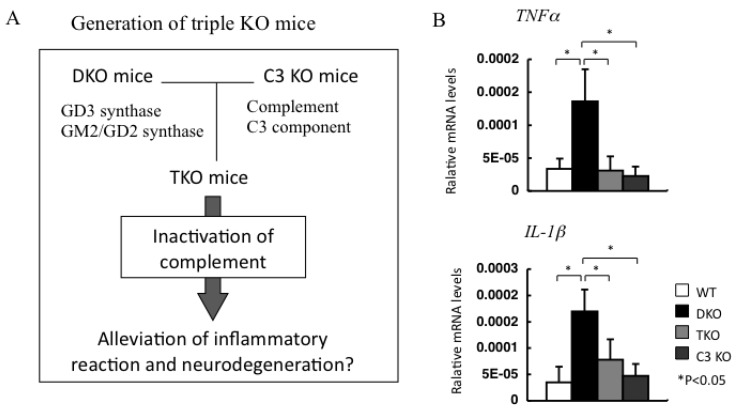
Inflammatory reactions in DKO mice were suppressed by genetic disruption of complement system. (**A**) Triple KO (TKO) mice were generated by mating DKO mice with C3 KO mice to clarify the roles of complement systems in brain disorders in DKO mice. (**B**) Expression levels of inflammatory cytokines were reduced in TKO mice. Relative mRNA levels of *TNF*α and *IL-1*β were compared using RT-qPCR.

**Figure 5 ijms-21-01906-f005:**
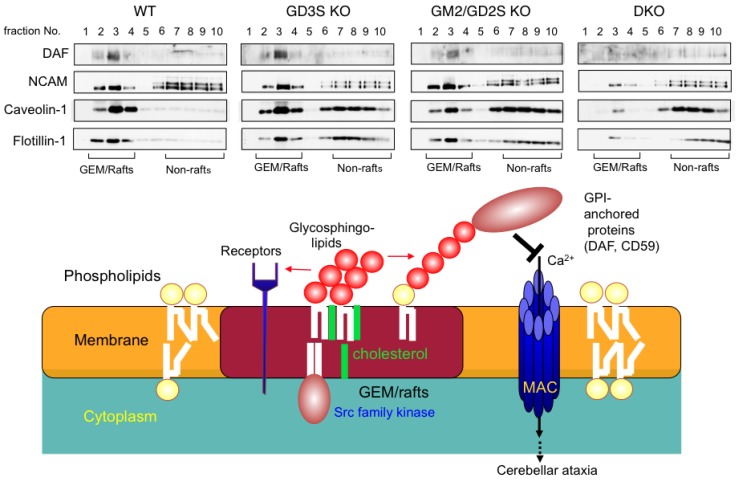
Destruction of the GEM/raft induced changes in the architecture and functions of raft-localizing molecules. Localization of GPI-anchored proteins including complement-regulatory molecules changed, leading to functional abnormalities and subsequent complement activation and inflammation in ganglioside-deficient mouse brain. GPI-anchored molecules such as DAF and NCAM as well as GEM/raft markers shifted from GEM/rafts to non-rafts domains. A part of a figure in Ref. 81 was presented after modification. Among isoforms of NCAM, only the GPI-anchored one was detected in GEM/rafts.

**Table 1 ijms-21-01906-t001:** Deficient structures, remaining glycolipids, and phenotypes of individual KO mice of glycosyltransferase genes. Residual glycolipids could compensate for the functions that deleted structures primarily exerted.

KO Gene	Glc-Cer Syn	GM3 Syn	GD3 Syn	GM2/GD2 Syn	DKO ^1)^
Lost structures	all glyco- sphingolipids	ganglio-series (a-, b-, c-)	b-series (and c-series)	all complex gangliosides(inc. asialo-series)	all complex gangliosides(inc. asialo-series)
Remaining structures		asialo-series	a-series and asialo-series	GM3, GD3 (and GT3)	GM3
	Gal-Cer and sulfatedes	Gal-Cer and sulfatedes neutral glycolipids	Gal-Cer and sulfatedes neutral glycolipids	Gal-Cer and sulfatedes neutral glycolipids	Gal-Cer and sulfatedes neutral glycolipids
Phenotypes	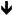	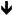	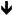	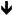	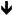
	Embryonal lethal	No apparent abnormalities	Mild abnormalities	Gradual abnormalities	Neurodegeneration from early phase
Remarks		Auditory disorder	Poor repair Low serum leptin	Male infertileity Low serum testostelone Wallerian degen ^2)^	Refractory skin lesion Auditory shock

^1)^, double KO of GD3 synthase and GM2/GD2 synthase genes; ^2)^, Wallerian degeneration.

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
