# Peer review of "Novel Molecular Mechanisms of Gangliosides in the Nervous System Elucidated by Genetic Engineering"

_ijms, 2020, doi:10.3390/ijms21061906_

Round 1
Reviewer 1 Report
The paper by Furukawa K et al., is a very interesting review with the title “Integrative roles of gangliosides in the nervous system: Novel molecular mechanisms elucidated by genetic engineering”. The authors focused on characterizing the molecular structure of gangliosides and its roles in the pathophysiological condition in the nervous system by using KO mice. The author has shown convincingly the role of gangliosides in diverse neurodegenerative diseases. However, in general, the figures are barely described. Also, Family members of AG important in the function of the nervous system in the introduction are bearly mentioned in the introduction.
Major comments
Interestingly, the authors describe the biological relevance of gangliosides in the physiological process (line 33-55) that are relevant for the neuronal pathophysiological process, suggesting that gangliosides might play an essential role in the neurological process. In the second paragraph, the authors described the roles of gangliosides types of gangliosides and their molecular structures, figure 1. In 2.1 is described as mature type gangliosides such as GMI, CD1a, GD1a, and GT1b. I will suggest describing the family members of gangliosides expressed in the nervous system and probably might highlight the biological relevance of almost all the members previously introduced. For example, in Alzheimer's diseases is mentioned GM1, GM2, and GD2 and GM3 metabolism is involved in the pathogenesis of AD. Are there KO animals for all of them??
Figure 2. The authors showed a schematic representation of the main pathway of ganglioside synthesis. Glycosyltransferases catalyzing individual steps were shown by italics, and deleted structures in KO mice were encircled by different colored squares. Authors need to clarify green letters correspond to different gangliosides.
Figure 3. Immunohistochemistry of GFAP-positive astrocytes and F4/80 antibody-reactive microglia. Marked gliosis was found in ganglioside synthase gene KO mice. GFAP-positive cells were increased (left), and F4/80-positive cells (right) were assembled in cerebella, indicating astrocytes and microglia, respectively. Authors need to clarify the meaning of GFAP and mentioned the read staining make the reference to ?? and green staining refer to??? There are yellow lines that correspond to the border in the transition of mol layer and grand layer. However, there are two yellow lines out of the figures. Need to clarify if authors are using for some structure or refer to??. The white scale bar needs to add size.
Figure 4. A, Triple KO (TKO) mice were generated by mating the DKO mice and C3 KO. Authors need to clarify the triple Ko mice correspond to which proteins??
Figure 5. Destruction of GEM/raft induced changes in the architecture and functions of raft- localizing molecules. Localization of GPI-anchored proteins including complement-regulatory molecules changed, leading to functional abnormalities and subsequent complement activation and inflammation in ganglioside-deficient mice brain. Authors need to clarify if that western blot corresponds to change of localization of GPI-anchored proteins and if these results are previously published, clarified in figure and add the reference.
Author Response
To the Reviewer 1:
Major comments:
In the second paragraph, the authors described the roles of gangliosides types of gangliosides and their molecular structures, figure 1. In 2.1 is described as mature type gangliosides such as GMI, CD1a, GD1a, and GT1b. I will suggest describing the family members of gangliosides expressed in the nervous system and probably might highlight the biological relevance of almost all the members previously introduced. For example, in Alzheimer's diseases is mentioned GM1, GM2, and GD2 and GM3 metabolism is involved in the pathogenesis of AD. Are there KO animals for all of them??
As the reviewer indicated, we did not explain roles of the individual gangliosides corresponding to Figure 1. Therefore, we have added brief explanation about major synthetic pathway of each series, and also mentioned about their involvement in Alzhermer’s disease in 2.1 (79-83). Since we described expression patterns of simple gangliosides and more complex gangliosides in early developmental stages and lager stages, respectively in 2.1 (74-78), These explanations (74-83) might be enough as introductory paragraphs for the main descriptions later. As for roles of the individual ganglioside structures, it seems hard to clearly define due to technical issues as we mentioned in Conclusion.
Figure 2. The authors showed a schematic representation of the main pathway of ganglioside synthesis. Glycosyltransferases catalyzing individual steps were shown by italics, and deleted structures in KO mice were encircled by different colored squares. Authors need to clarify green letters correspond to different gangliosides.
As the reviewer indicated, we used green letters without any explanation in Figure 2. Then, we have modified the color so that only names of glycolipids on the main pathway of ganglioside synthesis have been indicated by green letters, and it was described in the legend for Figure 1.
Figure 3. Immunohistochemistry of GFAP-positive astrocytes and F4/80 antibody-reactive microglia. Marked gliosis was found in ganglioside synthase gene KO mice. GFAP-positive cells were increased (left), and F4/80-positive cells (right) were assembled in cerebella, indicating astrocytes and microglia, respectively. Authors need to clarify the meaning of GFAP and mentioned the read staining make the reference to ?? and green staining refer to??? There are yellow lines that correspond to the border in the transition of mol layer and grand layer. However, there are two yellow lines out of the figures. Need to clarify if authors are using for some structure or refer to??. The white scale bar needs to add size.
As the reviewer indicated, we needed to explain the methods and results of Figure 3. First of all, we have exchanged the results of the immunohistochemistry of sections from 15-week- to 50-week-old mice, since these are more apparent without changing our conclusion. We have added descriptions on what is GFAP, information of antibodies and 2ndary reagents used for detection, and also on the scale bar in the legend for Figure 3. As for the yellow lines, we explained that they were drawn based on the results of Hematoxylin-Eosin staining of the serial section in the legend.
Figure 4. A, Triple KO (TKO) mice were generated by mating the DKO mice and C3 KO. Authors need to clarify the triple Ko mice correspond to which proteins??
As the reviewer indicated, we have added protein names deleted in DKO mice and C3 KO mice in Figure 4A. In addition, we have added explanations for B in the legend for Figure 4.
Figure 5. Destruction of GEM/raft induced changes in the architecture and functions of raft- localizing molecules. Localization of GPI-anchored proteins including complement-regulatory molecules changed, leading to functional abnormalities and subsequent complement activation and inflammation in gangliosidedeficient mice brain. Authors need to clarify if that western blot corresponds to change of localization of GPI-anchored proteins and if these results are previously published, clarified in figure and add the reference.
As the reviewer indicated, we have added description on the changes in the intracellular localization of GPI-anchored proteins, and also indicated the reference from which the immunoblotting data derived in the legend for Figure 5.
Reviewer 2 Report
The manuscript entitle “Integrative role of gangliosides in the nervous system…” by Koichi Furukawa, etal is a review article that recapitulated the role of gangliosides in the nervous system, focusing on their implication in neuroinflammation, aging and neurodegeneration. The authors of the review presented fairly convincing evidence that
congenital neurodynamic disorders are associated with mutations in the ganglioside synthase genes. For detection of functions of gangliosides they used ganglioside-deficient (knockout) mice.It is very interesting that inflammatory reaction was found mainly in the central nervous systems of many KO mice of glycosyltransferaseb genes. Of particular interest is the mechanism proposed by the authors about involvement of complement systems in neuro-inflammation as a novel aspect of ganglioside deficiency.
They compared degree of the complement activation, inflammation reaction and destribution of lipids rafts among varios KO mice of glycosyltransferase genes and wild tipe mice, and demonstrated increased expression level of complement-related genes.
The review includes quite a lot of cited works by other authors, but the authors' own articles are also presented in detail. In addition, figures are presented with the results obtained by the authors. Of the 131 references, 24 belong to the authors of the review. The review reflect good work the author have done. This information indicates that the authors are deeply familiar with the problem that is reflected in the review.
Some little remarks to correct in the text:
1.It is advisable to replace closely-spoken words.Line 19-higlyLine 20-high
2.Add a more detailed description of Fig. 3, as short text creates difficulties in understanding the obtained experimental data.
3.A number of references provide a complete list of authors of the article, and some are abbreviated. DOI is not indicated.
Author Response
To the Reviewer 2:
Some little remarks to correct in the text:
1.It is advisable to replace closely-spoken words.Line 19-higlyLine 20-high
As the reviewer suggested, we have changed the word “highly” to “predominantly”.
2.Add a more detailed description of Fig. 3, as short text creates difficulties in understanding the obtained experimental data.
As the reviewer indicated, we have added explanation on the conditions and reagents for immunohistochemistry, scale bars, and border lines in the legend for Figure 3 as responded to the reviewer 1’s comment.
3.A number of references provide a complete list of authors of the article, and some are abbreviated. DOI is not indicated.
As the reviewer indicated, the description of references was incomplete. We have corrected all references along with the journal’s rule.